No evidence for negative impacts of acute sulfoxaflor exposure on bee olfactory conditioning or working memory

Siviter Harry harry.siviter.2016@live.rhul.ac.uk
Scott Alfie
Pasquier Grégoire
Pull Christopher D.
Brown Mark J.F.
Leadbeater Ellouise
School of Biological Sciences, Royal Holloway University of London , Egham , UK
Anderson Todd
Electronic publication date: 2019 Aug 12
Publication date: 2019
Volume: 7
Electronic Location ID: e7208
Received 2019 Mar 18; Accepted 2019 May 29
Copyright: ©2019 Siviter et al.
Copyright year: 2019
Copyright holder: Siviter et al.
License: This is an open access article distributed under the terms of the Creative Commons Attribution License, which permits unrestricted use, distribution, reproduction and adaptation in any medium and for any purpose provided that it is properly attributed. For attribution, the original author(s), title, publication source (PeerJ) and either DOI or URL of the article must be cited.
License URL: https://creativecommons.org/licenses/by/4.0/

Keywords: Sulfoxaflor, Bumblebees, Neonicotinoid, Radial-arm maze, Spatial-working memory, Sulfoximine, Insecticide, Honeybee, Memory

Funding: Royal Holloway University European Research Council BeeDanceGap 638873 The Leverhulme Trust RGP-2016-444 European Horizon 2020 research and innovation programme 773921 Biotechnology and Biological Sciences Research Council BB/N000668/1 Harry Siviter was supported by a Royal Holloway University of London Reid PhD Scholarship and by contributions from High Wycombe Beekeeper’s Association. Elli Leadbeater’s contribution was supported by the European Research Council (Starting Grant BeeDanceGap 638873; honeybee section) and The Leverhulme Trust (RGP-2016-444; RAM section). This project has received funding from the European Horizon 2020 research and innovation programme under grant agreement no.773921 & Biotechnology and Biological Sciences Research Council, Grant/Award Number:BB/N000668/1. The funders had no role in study design, data collection and analysis, decision to publish, or preparation of the manuscript.

==============================
Systemic insecticides such as neonicotinoids and sulfoximines can be present in the nectar and pollen of treated crops, through which foraging bees can become acutely exposed. Research has shown that acute, field realistic dosages of neonicotinoids can negatively influence bee learning and memory, with potential consequences for bee behaviour. As legislative reassessment of neonicotinoid use occurs globally, there is an urgent need to understand the potential risk of other systemic insecticides. Sulfoxaflor, the first branded sulfoximine-based insecticide, has the same mode of action as neonicotinoids, and may potentially replace them over large geographical ranges. Here we assessed the impact of acute sulfoxaflor exposure on performance in two paradigms that have previously been used to illustrate negative impacts of neonicotinoid pesticides on bee learning and memory. We assayed whether acute sulfoxaflor exposure influences (a) olfactory conditioning performance in both bumblebees (Bombus terrestris) and honeybees (Apis mellifera), using a proboscis extension reflex assay, and (b) working memory performance of bumblebees, using a radial-arm maze. We found no evidence to suggest that sulfoxaflor influenced performance in either paradigm. Our results suggest that despite a shared mode of action between sulfoxaflor and neonicotinoid-based insecticides, widely-documented effects of neonicotinoids on bee cognition may not be observed with sulfoxaflor, at least at acute exposure regimes.

Introduction

Bees provide vital pollination services for both wild flowers and commercial crops (Rader et al., 2016; Fijen et al., 2018), so localised declines in domestic honey bee populations and both localised and global range reductions of certain bumblebee species have led to suggestions that a global pollination crisis could be imminent (Biesmeijer et al., 2006; Colla & Packer, 2008; Aizen & Harder, 2009; Williams & Osborne, 2009; Potts et al., 2010; Cameron et al., 2011; Kerr et al., 2015; Goulson et al., 2015). Although the intensification of agriculture, habitat loss, global warming and pathogen exposure have all been linked with bee declines (Brown & Paxton, 2009; Winfree et al., 2009; Cameron et al., 2011; Kerr et al., 2015; Goulson et al., 2015; Samuelson et al., 2018), particular attention has focused on the impact of agrochemicals. A key focus of research has been to understand the impact of neonicotinoid-based insecticides on bees (Whitehorn et al., 2012; Godfray et al., 2014; Godfray et al., 2015; Stanley et al., 2015a; Rundlöf et al., 2015; Goulson et al., 2015; Kessler et al., 2015; Woodcock et al., 2016; Woodcock et al., 2017; Baron, Raine & Brown, 2017b; Baron et al., 2017a; Tsvetkov et al., 2017; Main et al., 2018; Arce et al., 2018; Siviter et al., 2018b), leading to controversy worldwide and in some cases, legislative reassessment of their use. Sulfoximine-based insecticides, which share a mode of action with neonicotinoids as selective agonists of Nicotinic Acetyl Choline Receptors (NAChRs) (Zhu et al., 2011; Sparks et al., 2013), are a more recent entry to the insecticide market, and are currently approved for use in 81 countries around the world. In a recent horizon-scanning exercise involving 72 pollination biologists, sulfoximines were highlighted as an emerging potential threat to pollinators, based on a lack of knowledge regarding their sub-lethal effects (Brown et al., 2016).

Sulfoxaflor, the first branded sulfoximine-based insecticide, can negatively impact bumblebee colony fitness, reducing worker production and subsequent reproductive output (Siviter, Brown & Leadbeater, 2018a), and so the effects are comparable to those observed with neonicotinoids (Whitehorn et al., 2012; Rundlöf et al., 2015). A plethora of research on neonicotinoids has linked small sub-lethal effects on bee behaviour at the individual level to major impacts at the colony level, with neonicotinoid exposure influencing bee foraging success and motivation, (Gill, Ramos-Rodriguez & Raine, 2012; Feltham, Park & Goulson, 2014; Gill & Raine, 2014; Arce et al., 2017; Lämsä et al., 2018; Muth & Leonard, 2019), homing success (Henry et al., 2012; Fischer et al., 2014), brood care and thermoregulation (Crall et al., 2018). One way in particular that neonicotinoids may influence bee behaviour is through impacts on bee cognition, and a recent meta-analysis has confirmed the detrimental effects of insecticide exposure on learning and memory at acute and field realistic regimes (Siviter et al., 2018b). As a systemic insecticide, sulfoxaflor, like neonicotinoids, can be present in the nectar and pollen of plants following treatment, meaning that foraging bees may be exposed either via the crop itself or through flowering weeds present in fields or orchards during spray (Botias et al., 2015; Kyriakopoulou et al., 2017). However, despite the similarity in mode of action between sulfoxaflor and neonicotinoids, the potential impact of sulfoxaflor exposure on bee learning and memory has not been tested.

In this study, we assay the impact of acute sulfoxaflor exposure on learning and memory in bees based on two paradigms through which previous authors have identified adverse effects of neonicotinoid exposure: a Proboscis Extension Reflex (PER) experiment (Stanley, Smith & Raine, 2015b; Siviter et al., 2018b) and a Radial Arm Maze-based assay (RAM; Samuelson et al., 2016). These paradigms are assays of (i) classical conditioning of olfactory stimuli and (ii) working memory (also known as short-term memory) respectively, and thus they may capture different aspects of foraging, although are unlikely to be mutually exclusive. For example, learning to discriminate between olfactory stimuli in a PER task may emulate learning to discriminate between rewarding and non-rewarding flower species, while RAM performance assays retention of short-term task-relevant information such as the location of flowers that a bee has recently visited (Foreman & Ermakova, 1998; Lihoreau, Chittka & Raine, 2010; Collett, Chittka & Collett, 2013; Samuelson et al., 2016). Exposure to certain neonicotinoids, and other non-neonicotinoid insecticides, has been shown to influence PER performance in both Apis and Bombus (Williamson, Baker & Wright, 2013; Stanley, Smith & Raine, 2015b; Piiroinen & Goulson, 2016; Siviter et al., 2018b), while impacts on RAM performance have only been tested in bumblebees (Samuelson et al., 2016). Given that sulfoxaflor and neonicotinoids both act as agonists of nicotinic acetylcholine receptors (NAChRs) (Sparks et al., 2013), we predicted that sulfoxaflor exposure would have similar negative impacts on PER performance in Apis and Bombus, and RAM performance for Bombus.

Methods: PER—Experiment 1

Subjects and harnessing

Five bumblebee (Bombus terrestris audax) colonies, each with approximately 150 workers, were purchased (Koppert Ltd, Haverhill, UK) and moved into wooden colony boxes (28 × 10 × 18 cm) connected to flight arenas (100 × 70 × 50 cm) that contained an ad libitum supply of sucrose solution (50° Brix) and pollen. Only individuals that had been observed foraging on the feeder within flight arenas were subsequently used in the experiment (Martin, Fountain & Brown, 2018). Previous studies suggest that bumblebees are more responsive when starved for a period of time (Stanley, Smith & Raine, 2015b), and consequently prior to all PER experiments involving bumblebees, we collected and harnessed all potential subjects before leaving them overnight, and testing them the following morning.

Returning foraging honey bees (Apis mellifera) were collected from the entrance of five hives from a research apiary at Royal Holloway University of London. Honeybee mortality is high when individuals are harnessed for a sustained period of time, and as a result we collected and harnessed honeybees in the same day, leaving them for one hour after harnessing, before randomly assigning them to a treatment group (see below) and conducting the experiment. Bumblebees and honeybees were tested on different days, and on any single test day, sixteen to forty bumblebees and honeybees were collected and harnessed.

Insecticide exposure

Sulfoxaflor has been developed for a range of different crops, including as a seed treatment for bee attractive crops, but its most common application is currently as a foliar spray (Centner, Brewer & Leal, 2018). Foliar spray applications result in short-term bursts of high insecticide residues in the nectar of sprayed crops (United States Environmental Protection Agency, 2016) and any concurrently flowering weeds. We thus based our estimates for acute exposure on data for the residue levels found in honeybee-collected nectar of cotton sprayed with sulfoxaflor from an Environmental Protection Agency (EPA) study, which demonstrated that over an 11 day period nectar concentrations ranged from 5.41–46.97 ppb (United States Environmental Protection Agency 2016; application rate: 0.045 pounds (0.020 kg) of active ingredient per acre, applied twice). We derived our sulfoxaflor treatments from a stock solution of 1 g dm−3 sulfoxaflor (Greyhound Chromatography and Allied Chemicals) in acetone, which was combined with sucrose solution (50° Brix) to make three treatment groups: 2.4 µg dm−3 (2.4 ppb), 10 µg dm −3 (10 ppb), 250 µg dm−3(250 ppb; positive control) and the negative control (sucrose with acetone only). Before training, we placed each bee horizontally (held in place with modelling clay) and pipetted a 10µl droplet of sucrose solution containing the randomly-assigned treatment solution onto a plastic surface, from which the bees could feed. Bees that did not immediately drink were encouraged to extend their proboscis by antennal stimulation with sucrose. Bees that did not consume the full quantity of sucrose solution were not used in the experiment (excluded bumblebees N = 55, control = 13, 2.4ppb = 16, 10 ppb = 16, 250 ppb = 10; honeybees N = 17, control = 2, 2.4 ppb = 6, 10 ppb = 5, 250 ppb = 4). After feeding, the bees were placed upright and left for an hour (Stanley, Smith & Raine, 2015b).

Training protocol

We used an absolute conditioning proboscis extension reflex (PER) procedure in which lavender scent (conditioned stimulus; CS) was forward paired with antennal stimulation by sucrose solution (unconditioned stimulus; US; 50° Brix). The subjects were placed three cm away from the odour tube that contained filter paper soaked in 4 µl of the lavender essential oil. A programmable logic controller computer was used to blow a constant stream of air containing the odours towards the subjects from the odour tube. The odour tube was replaced every 20–30 trials to ensure that the odour was consistently strong throughout conditioning. Bees were exposed to 5 s of clean airflow (no odour), followed by 10 s of the odour. Six seconds after the start of odour exposure, the subject was presented with 0.8 µl of untreated sucrose solution (50° Brix) from a syringe. A positive response was recorded if the bee extended its proboscis in the first six seconds of odour presentation, before antennal stimulation with the US, and was always rewarded by immediate delivery of the sucrose solution. In the event of a negative response, we additionally recorded whether the bee responded to the antennal stimulation (to ascertain that the subject was motivated to extend its proboscis). Each subject received fifteen trials with an inter-trial interval of approximately 12 min. To ensure that the bees were learning about the odour and not other aspects of the experimental protocol, three non-scented probe trials were randomly distributed between the 5th and 15th learning trials. Bees that responded to the unscented stimulus in any probe trial were not included in the analysis (excluded bumblebees n = 10; honeybees n = 1). Each animal thus received 18 trials in total (15 test trials and 3 probe trials).

Medium- and long-term memory tests, whereby the subjects were presented with the conditioned odour in isolation for a single trial, were conducted with the same subjects 3 h and 24 h after the last learning trials, respectively. Once the experiment was finished, bees were frozen and their size recorded by measuring thorax width with electronic callipers (Mitutoyo), three times, from which a mean value was taken. We recorded size because it may influence the rate at which the insecticide is absorbed; larger bees empty their gut at a faster rate (Fournier et al., 2014) and previous studies have correspondingly found size-dependent effects of acute insecticide exposure on cognition (Samuelson et al., 2016).

In total, we tested 240 bumblebees and 174 honeybees. Bees that did not extend their proboscis in response to antennal sucrose stimulation in at least 5 learning trials were not used (bumblebee N = 64, control = 17, 2.4 ppb = 14, 10 ppb = 16, 250 ppb = 17; honeybee N = 6, control = 1, 2.4 ppb = 1, 10 ppb = 2, 250 ppb = 1). A further 3 bumblebees were removed from the experiment because they extended their proboscis before the odour was presented. Five bumblebees died, as did 46 honeybees. One bumblebee was harnessed poorly and thus not included, as were 10 honeybees. This resulted in final sample sizes of 102 bumblebees and 94 honeybees (bumblebees: control = 23, 2.4 ppb = 26, 10 ppb = 24, 250 ppb = 29: honeybees: control = 29, 2.4 ppb = 22, 10 ppb = 22, 250 ppb = 21).

Statistical analysis

We followed an information theoretic model selection approach. The initial model set included a full model and all subsets, including a null model that contained solely the intercept and “Colony” as a random factor. We selected a 95% confidence set of models based on Akaike weights derived from AICc values. In cases where the 95% confidence set contained more than one model, the models were averaged (Burnham & Anderson, 2002) (including the null if it was included within the confidence set) to produce parameter estimates and 95% confidence intervals. Data collected for bumblebees and honeybees were analysed separately due to potential differences between the species (see Siviter et al., 2018b).

Following Stanley, Smith & Raine (2015b), we analysed three dependent variables to identify sulfoxaflor effects on PER performance: (i) whether the bee responded to the CS in the absence of antennal stimulation (hereafter: “positive response”) in at least one trial overall (ii) the total number of positive responses (hereafter learning level) from bees that learnt the association, and (iii) the trial that the bee first exhibited a positive response. We used generalised linear mixed effect models with binomial or Poisson error structures, or mixed effect Cox models, respectively, where treatment, bee size and their interaction were specified as fixed factors, and colony as a random factor (see Tables S1 & S2). For medium- and long-term memory, we analysed whether or not the bee exhibited a positive response to the CS following the same method (binomial error structure). We used the packages, lme4 (Bates et al., 2015), MuMin (Barton, 2016), Coxme (Therneau, 2018), Hmisc (Harrell & Dupont, 2018) and pscl (Jackman, 2017).

Methods: RAM—Experiment 2

Subjects

Seven bumblebee colonies (B. terrestris audax), each with approximately 150 workers, were obtained from Biobest (Agralan Ltd, Swindon, Wiltshire, UK) and upon arrival each was transferred into a plastic bipartite nest box (28 × 16 × 10.5 cm, with a central divider that allowed access between compartments). When transferring bees into the nest box individual bees were tagged with unique number disks, allowing the identification of individuals. During experiments, the nest box was attached to the radial arm maze (RAM; description below), with access controlled using sliding trap-doors. When the bees were not being tested, gravity feeders were placed in the RAM with an ad libitum supply of 43° Brix sucrose solution. Colonies were provided with approximately 7 grams of pollen in the nest box 3 times a week. Colonies were used in succession rather than simultaneously, and newly emerged bees were tagged daily throughout the experiment.

Radial arm maze

A radial arm maze classically consists of 8 arms, each of which contains a hidden food reward (Foreman & Ermakova, 1998). Animals forage within the maze and search for the food rewards, whilst avoiding re-visiting arms that they have already depleted, and Samuelson et al. (2016) have previously confirmed that bumblebees use working memory to minimize such revisits. We based our design on the set-up used by Samuelson et al. (2016) but modified their original vertical design to create a horizontal version. The aim of this modification was to reduce reliance on learnt movement rules by forcing subjects to return to the centre of the maze between choices, as is usually the case for rodent versions of the RAM (Olton & Samuelson, 1976; Foreman & Ermakova, 1998). Our horizontal maze was constructed from acrylic plastic, sealed together with non-toxic grey silicone (Bondit). Each of the 8 arms contained a removable platform (7.2 × 2.6 × 0. five mm) upon which the bees could land to access a small hole in the wall. By extending the proboscis through this hole, bees could access a sucrose reward (43° Brix) that was not visible from the platform (volume varied between stages; see below). After visitation, the platform could be rapidly replaced with a clean alternative to prevent the use of scent marks to identify visited arms. The availability of visual global landmarks (often a view of the laboratory) has been shown to contribute to performance in a RAM for rodents and other animals (Foreman & Ermakova, 1998; Wilkinson, Chan & Hall, 2007), but (a) our laboratory regularly changes in appearance and (b) light control was important for our video software. Thus, our maze walls were opaque, but papered with a black and white panoramic photo of the laboratory to allow bees to orientate.

Stage 1—Group training

The objective of this stage was to identify motivated foragers. Each morning before testing, the RAM was set up with 10 µl 43° Brix sucrose solution on each landing platform. All bees were then allowed into the RAM to forage on the landing platforms (platforms were continuously reloaded with sucrose solution when drained). Only bees that were observed foraging within the maze at this stage (by inserting the proboscis into the holes at the end of each arm) proceeded to Stage 2 (Individual training).

Stage 2—Individual training

The objective of the individual training stage was for bees to learn the win-shift nature of the RAM task, over the course of 10 training bouts. During each bout, bees were required to visit all eight artificial platforms and then return to the nest box to empty their crop. At the onset of each bout, each platform contained 10 µl of sucrose solution (20 µl for the first bout, to increase motivation). Rewards were not refilled after visitation, but landing platforms were replaced with identical but clean replacements. Once the bee found the final reward, we increased the amount of sucrose solution in that arm (from outside the maze) so that the bee’s crop was full, encouraging her to return to the nest box. Choices in the RAM were recorded as either: Correct –feeding from platforms that had not yet been visited, or Incorrect- attempting to feed from platforms that had already been depleted.

If a bee attempted to return to the nest box three times prior to visiting all landing platforms, or if the trial exceeded 20 min, the bee was permitted to return to the nest box. As with Samuelson et al. (2016), each bee completed 10 training bouts.

Stage 3—Pesticide exposure

Our pesticide exposure regime differed from that used in the PER regime because our RAM experiment was designed to allow direct comparison with the results described by (Samuelson et al., 2016) for thiamethoxam. Samuelson et al. (2016) aimed to mimic the dosage received during one hour of foraging for nectar, whilst overcoming the problem that feeding on a large volume of sucrose may reduce a bee’s motivation to participate in the RAM. To that end, bees were only fed half of the volume of nectar that would normally be consumed during such a foraging bout (0.5 × 37.7 mg), with a doubled concentration of sulfoxaflor. To allow for direct comparison, we followed the same approach here (and bumblebees thus received a higher dosage than those in the PER treatment groups described above). Each test subject was intercepted as it was returning to the RAM after emptying its crop following the 10th training bout. They were placed into a plastic beaker, and fed 18.85 µl of sucrose solution from the randomly assigned treatment group. We included four treatment groups, intended to mimic foraging on crops with nectar containing either 0 ppb (control), 5 µg dm−3 (5 ppb), 10 µg dm−3 (10 ppb) or 250 µg dm−3 (250 ppb or positive control) of sulfoxaflor, so bees from each treatment group received 0, 0.045, 0.091 & 2.5 ng respectively. After consumption, the bees were held in the plastic beaker for 45 min before being returned to the nest (Samuelson et al., 2016). 60 bees were originally trained on the RAM but 2 failed to re-commence foraging after the exposure stage (N values, control = 14, 5 ppb = 15, 10 ppb = 15, 250 ppb n = 14).

Stage 4—Test trial

Following exposure, the bees were presented with the exact set up they had experienced in the training phase of the experiment (phase 2) and tested one final time. After completing the task bees were collected and frozen and, at a later date, we measured their thorax width.

Statistical analysis

As with experiment 1, we used an information theoretic model selection approach when analysing each dependent variable and, as in previous work (Olton & Samuelson, 1976; Foreman & Ermakova, 1998; Samuelson et al., 2016), three different measures were chosen to assess performance; (i) total revisits to platforms which have been previously visited, (ii) the number of correct choices made before making a revisit and (iii) the proportion of correct choices in the first eight visits. For all dependent variables, treatment, bee size and the interaction between them were included as fixed factors with colony included as a random factor. To account for overdispersion, we used a generalised linear model with a negative binomial distribution error structure (glm.nb) to analyse total revisits, and a generalised linear model (glm) with a Poisson distributed error structure to analyse the number of correct choices in the first 8 visits. A mixed effect cox model (coxme) was used to analyse correct choices before first revisit. All analyses were conducted in R studio (version 1.1.419) using the R packages lme4 (Bates et al., 2015), MuMin (Barton, 2016), Coxme (Therneau, 2018), AER (Kleiber & Zeileis, 2008), MASS (Ripley & Venables, 2002), Hmisc (Harrell & Dupont, 2018).

Results: PER—Experiment 1

For our first measure of learning (production of at least one conditioned response to the stimulus), we found no evidence that acute sulfoxaflor exposure influenced bumblebees (Fig. 1A, glmer, 2.4 ppb parameter estimate (PE) = −0.00, 95% CI [−0.34 to 0.33]; 10 ppb PE = 0.00, 95% CI [−0.35 to 0.36]; 250 ppb PE = 0.05, 95% CI [−0.43 to 0.53]) or honeybees (Fig. 2A; glmer, 2.4 ppb PE = −1.30, 95% CI [−14.19 to 11.60; 10 ppb PE = −1.26, 95% CI [−14.82 to 12.31]; 250 ppb PE = −7.32, 95% CI [−53.10 to 38.45]). Learning level (number of positive responses) was also not influenced by sulfoxaflor exposure (bumblebees, Fig. 1B; glmer; wi (treatment) = 0.017; honeybees, Fig. 2B; glmer, 2.4 ppb PE = 1.18, 95% CI [−8.23 to 10.59]; 10 ppb PE = 1.05, 95% CI [−6.93 to 9.04]; 250 ppb PE = 0.31, 95% CI [−4.11 to 4.72]). Finally, there was no evidence to suggest that sulfoxaflor exposure influenced the speed at which either bumblebees or honeybees learnt the olfactory association (bumblebees, Fig. 1C, coxme, 2.4 ppb PE = −0.00, 95% CI [−0.93 to 0.78]; 10 ppb PE = −0.00, 95% CI [−0.91 to 0.87]; 250 ppb PE = 0.03, 95% CI [−0.39 to 1.22]; honeybees, Fig. 2C; coxme, 2.4 ppb PE = −0.11, 95% CI [−0.72 to 0.51]; coxme, 10 ppb PE = −0.02, 95% CI [−0.34 to 0.29]; coxme, 250 ppb PE = −0.01, 95% CI [−0.30 to 0.28]), suggesting no influence of acute sulfoxaflor exposure on olfactory conditioning performance in either species.

Figure 1 Bumblebee olfactory learning.

The performance of bumblebees in an olfactory learning task (A) the proportion (±SEM) of bumblebees that learnt the olfactory association (B) the learning level (±SEM) of the bees that did learn the association and (C) the trials in which bees learnt the association (±SEM) in reference to trial number. (Control n = 23, 2.4 ppb n = 26, 10 ppb n = 24, 250 ppb n = 29).

Figure 2 Honeybee olfactory learning.

The performance of honeybees in an olfactory learning task: (A) The proportion (±SEM) of honeybees that learnt the olfactory association (B) the learning level (±EM) of the bees that did learn the association (B) and (C) the trials in which bees learnt the association (±SEM) in reference to trial number. (Control n = 29, 2.4 ppb n = 22, 10 ppb n = 22, 250 ppb n = 21).

Similarly, there was no impact of sulfoxaflor exposure on either bumblebee or honeybee memory at 3 h after training (bumblebee; Fig. 3A; glmer, 2.4 ppb PE = 0.02, 95% CI [−0.59 to 0.63]; 10 ppb PE = −0.07, 95% CI [−0.98 to 0.83]; 250 ppb PE = 0.06, 95% CI [−0.62 to 0.75]; honeybee; Fig. 3C; wi (treatment) = 0.033) or at 24 h after training (bumblebee; Fig. 3B; wi (treatment) = 0.042; honeybee ; Fig. 3D; glmer, 2.4 ppb PE = −0.39, 95% CI [−1.79 to 1.02]; 10 ppb PE = −0.36, 95% CI [−1.66 to 0.94]; 250 ppb PE = 0.04, 95% CI [−0.79 to 0.88]).

Figure 3 Figure 3: Bumblebee and honeybee olfactory memory.

The proportion of bumblebees and honeybees (±SEM) responding to the conditioned stimuli 3 hours (A & B) and 24 hours (C & D) after training had finished. (Bumblebee 3H, Control n = 10, 2.4 ppb n = 12, 10 ppb n = 11, 250 ppb n = 17; bumblebee 24H Control n = 9, 2.4 ppb n = 11, 10 ppb n = 9, 250 ppb n = 14; Honeybee 3H, Control n = 28, 2.4 ppb n = 21, 10 ppb n = 22, 250 ppb n = 20; honeybee 24H Control n = 23, 2.4 ppb n = 13, 10 ppb n = 17, 250 ppb n = 16).

Results: RAM—Experiment 2

We found no statistical support for an effect of sulfoxaflor exposure on total revisits (Fig. 4A; glm.nb, 5 ppb treatment PE = 0.24, 95% CI [−0.56 to 1.05]; 10 ppb PE = 0.16, 95% CI [−0.46 to 0.79]; 250 ppb PE = 0.23, 95% CI [−0.55 to 1]) or on the proportion of correct choices in the first 8 visits of bumblebees following sulfoxaflor exposure (Fig. 4B glm, (wi (treatment) = 0.038). Similarly, we found no statistically significant effect of sulfoxaflor exposure on the number of correct choices before the first revisit (Fig. 5; coxme, 5ppb PE = 0.55, 95% CI [−0.54 to 1.64]; 10 ppb PE = 0.25, 95% CI [−0.48 to 0.98]; 250 ppb PE = 0.49, 95% CI [−0.52 to 1.51]), suggesting no impact of acute sulfoxaflor exposure on bumblebee working memory. Further analysis also suggested no impact on bumblebee behaviour (see Supplemental Information).

Figure 4 Figure 4: Bumblebee performance on the radial arm maze.

(A) the total number of revisits (±SE) to already depleted landing platforms and, (B) the mean number of correct landings (±SE) in the bees first 8 landings.

Figure 5 Kaplan–Meier curves of bumblebees visiting landing platforms until a revisit to an already depleted resource occurs.

A, Control (n = 14); B, 5ppb (n = 15); C, 10 ppb (n = 15) and D, = 250 ppb (n = 14).

Discussion

We found no evidence to suggest that acute sulfoxaflor exposure influenced bumblebee or honeybee olfactory conditioning or bumblebee working memory, even at the highest concentrations of exposure tested (250 ppb). Given the range of dosages we tested, which included positive controls that far exceeded levels likely to be found in the field, it is unlikely that acute sulfoxaflor exposure in adult bees will influence cognition after environmental exposure, at least with regard to olfactory conditioning and working memory performance.

We used two experimental paradigms to investigate the impact of acute sulfoxaflor exposure on bee learning and memory. Although a wide variety of different paradigms can be used to assess bee cognition (Bernadou et al., 2009; Zhang & Nieh, 2015; Lämsä et al., 2018; Muth, Francis & Leonard, 2019) we chose to use both PER and the RAM, in combination, as these paradigms allow us to consider the impact of sulfoxaflor exposure on working memory (also known as short-term memory), medium-term and long-term memory (Menzel, 2012). Interestingly, in both of these paradigms, the neonicotinoid thiamethoxam, one of the three neonicotinoids insecticides banned from outdoor agricultural use within the European Union, has been shown to influence performance at comparable dosages (Stanley, Smith & Raine, 2015b; Samuelson et al., 2016). Both neonicotinoids and sulfoximine-based insecticide share the same mode of action, acting as selective agonists of Nicotinic Acetyl Choline Receptors (NAChRs) (Zhu et al., 2011; Sparks et al., 2013). Acute neonicotinoid exposure can inactivate the mushroom bodies of bee brains (Palmer et al., 2013), which are essential for learning and memory in bees (Menzel, 2012). The effects of sulfoxaflor exposure on bee neurology have not been explored, but could provide useful information in understanding why neonicotinoids, but not sulfoximine-based insecticides, influence bee cognition, at least under these experimental paradigms and dosages. Ultimately, sulfoxaflor could be used as a reference substance to understand why some insecticides, which act on nicotinic acetyl choline receptors (NAChRs) have a negative impact on bee cognition, while others do not.

We tested the impact of acute sulfoxaflor exposure (rather than chronic exposure) on bee learning and memory. A recent meta-analysis showed that chronic insecticide exposure can have larger effects on bee memory than acute exposure for adult bees, and so we cannot rule out that more prolonged exposure would have identified an effect of sulfoxaflor exposure. However, an acute dosage potentially mimics the exposure regime of a foraging adult bee in the field more closely, because individuals may forage on a range of different crops and flowers in addition to the treated crop, over an extended period of time. Chronic exposure is nonetheless clearly relevant for larval brood, and the same meta-analysis highlighted that exposure as a larva is more likely to have a negative impact on bee learning than adult exposure (Siviter et al., 2018b). Larval exposure to thiamethoxam has been shown to influence synaptic density in the mushroom bodies of bee brains (Peng & Yang, 2016) and increase neural vulnerability to mitochondrial dysfunction (Moffat et al., 2015), which may be linked to documented effects of exposure on cognitive function (Klein et al., 2017). Thus, although our results show no effect of acute sulfoxaflor exposure on bumblebee or honeybee cognition, further research needs to be conducted to understand the potential impact of chronic exposure, both in adults and larvae. Furthermore, given the dearth of data on non-Apis/Bombus bees (Siviter et al., 2018b), researchers should prioritise assessing the impact of sulfoxaflor on non-social bees.

The hypothesis that negative effects of neonicotinoid exposure on bees are mediated in part by the widely-documented sub-lethal effects on learning and memory described above, which may impact upon bee foraging behaviour and thus potentially colony productivity, has received much attention (Klein et al., 2017; Siviter et al., 2018b). However, neonicotinoid insecticides have many other sub-lethal effects on bee behaviour and physiology (Wu et al., 2012; Laycock et al., 2012; Baron, Raine & Brown, 2017b; Baron et al., 2017a; Crall et al., 2018) and any causal link between reduced cognitive performance and foraging efficiency remains to be established, because data linking bee cognitive traits and foraging efficiency are difficult to collect. The evidence that does exist is contradictory. Raine & Chittka (2008) found a positive association between the nectar collection rate of workers allowed to forage outdoors, and the visual learning performance of their sisters from the same colony, but Evans, Smith & Raine (2017) found no correlation between individual visual learning performance and nectar collection rate. A better understanding of the relationship between bee cognitive traits and foraging efficiency is clearly important if we are to identify and mitigate against the sub-lethal impacts that underlie negative impacts of neonicotinoid insecticide exposure on bumblebee colony reproductive output (Whitehorn et al., 2012; Rundlöf et al., 2015; Woodcock et al., 2017). In contrast, our findings suggest that sub-lethal effects on learning and memory are unlikely to underlie the negative impacts of sulfoxaflor on colony reproductive output in bumblebees.

If memory and learning are unaffected by exposure, what other mechanisms might underlie the impact of sulfoxaflor on bumblebee colony fitness (Siviter, Brown & Leadbeater, 2018a)? While previous work on impacts of neonicotinoids on learning and memory (Samuelson et al., 2016; Siviter et al., 2018b) inspired the work reported here, these insecticides have been demonstrated to produce a range of sublethal impacts, beyond cognitive effects. These include reductions in food intake, foraging motivation, thermoregulatory activity, nursing behaviour, ovary development, and egg laying (Laycock et al., 2012; Baron, Raine & Brown, 2017b; Lämsä et al., 2018; Crall et al., 2018). Impacts of sulfoxaflor on bumblebee colony fitness appear to be driven by reduced worker production at the early stage of colony development (Siviter, Brown & Leadbeater, 2018a), but our results here suggest that this is unlikely to be due to impacts on worker learning or memory in food-related tasks. Consequently, we suggest that future work should focus on examining potential sub-lethal impacts on ovary development and egg laying, which could directly relate to reductions in worker production.

Conclusions

Sulfoximine-based insecticides are becoming globally important, and sulfoxaflor is now registered for use in 81 countries, including a number of European Union member states (European Commission, 2018). Although mitigation measures can reduce the likelihood of pollinator exposure (Centner, Brewer & Leal, 2018), uptake of such measures varies widely across legislative regimes. Previous work with neonicotinoids demonstrated the importance of understanding sub-lethal effects of insecticides on bee health. Here we find no evidence for an impact of acute sulfoxaflor exposure on bee olfactory conditioning or bumblebee working memory, despite the occurrence of such impacts when using the same protocols with neonicotinoid exposure. This suggests that the impacts of sub-lethal exposure in learning and memory are unlikely to be the mechanism behind impacts of sulfoxaflor on colony reproductive success (Siviter, Brown & Leadbeater, 2018a). Further studies are needed to understand how, and under what conditions, sulfoxaflor may impact bee health. Such data will enable more informed regulatory and policy decisions on the future use of this insecticide in crops that attract bees.

Supplemental Information

Table S1 Candidate models for each analysis

Click here for additional data file.

Table S2 Parameter estimates and 95% confidence intervals derived by model averaging across the confidence set of models

Parameters highlighted in bold have 95% confidence intervals that do not cross zero

Click here for additional data file.

Supplemental Information 1 Bumblebee PER data

Click here for additional data file.

Supplemental Information 2 RAM raw data

Click here for additional data file.

Supplemental Information 3 Honeybee PER data

Click here for additional data file.

Supplemental Information 4 Supplementary information

Supplementary test: Does sulfoxaflor influence bee behaviour?

Click here for additional data file.

We would like to thank Ash Samuelson and Callum Martin for offering methodological advice.

Additional Information and Declarations

Competing Interests

Author Contributions

Data Availability

The authors declare there are no competing interests.

Harry Siviter conceived and designed the experiments, performed the experiments, analyzed the data, contributed reagents/materials/analysis tools, prepared figures and/or tables, authored or reviewed drafts of the paper, approved the final draft.

Alfie Scott performed the experiments, authored or reviewed drafts of the paper, approved the final draft.

Grégoire Pasquier and Christopher D. Pull contributed reagents/materials/analysis tools, approved the final draft.

Mark J.F. Brown and Ellouise Leadbeater conceived and designed the experiments, authored or reviewed drafts of the paper, approved the final draft.

The following information was supplied regarding data availability:

Siviter, Harry. 2019. “No Evidence for Negative Impacts of Acute Sulfoxaflor Exposure on Bee Olfactory Learning or Working Memory.” OSF. July 18. https://osf.io/ax8tv/.

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
