# Peer review of "No evidence for negative impacts of acute sulfoxaflor exposure on bee olfactory conditioning or working memory"

_PeerJ, doi:10.7717/peerj.7208_

## Round 0.1 · original submission · Minor Revisions

Please consider all of the reviewer comments and revise your manuscript accordingly.

Reviewer 1 ·

Basic reporting

The work presented by Siviter et al. titled " No evidence for negative impacts of acute sulfoxaflor exposure on bee olfactory conditioning or working memory" is of great interest and fits well the scope of the journal. In my regard, the manuscript is well structured. Since English is my second langue, I don’t have any comment on writing of this manuscript.

Experimental design

The methods are adequate and the results clearly presented.

Validity of the findings

Author found that not all neonicotinoids have the same effects on bee cognition. The result showed that other mechanisms might underlie the impact of sulfoxaflor on bumblebee colony fitness. It also reminded us more indexes should be accounted in evaluating impact of sulfoxaflor or next generation neonicotinoids on non-target organism.

Additional comments

No comments

Reviewer 2 ·

Basic reporting

No comment

Experimental design

No comment

Validity of the findings

No comment

Additional comments

I think these two experiments are well thought-out, thorough and highly appropriate for publication in PeerJ. My two main queries would be whether the authors measured other behavioural measures in experiment 2 that might be affected by the pesticide aside from learning, for example time spent foraging or movement patterns? It seems like a rich data set and so it would be interesting to know if any behaviour at all seems to be affected by the treatment. If other measures have been addressed, it might be worth including this, even just in the supplementary material. My second main query is about whether another statistical test could have been used in the first experiment (details below). Other smaller comments are listed below.

Line 94: Remove comma after “acute”
Line 98-99: isn’t the RAM not only a measure of working memory, but specifically spatial working memory? I’d be specific here, since it seems that STM could also be captured in the PER assay.
Line 107: “hypothesised” should be “predicted”
Line 107-109: it would be good to hear more about the specific mode of action here and why this is thought to affect learning performance
Line 115: 50% sucrose by weight or by volume? – I think Brix is usually reported as degrees Brix rather than a percentage?
Line 156: was the sucrose reward offered to the bees via a pipette?
Line 169: were all three size measures used or was an average taken? Also I think it would be good to mention why you’re measuring bee size and how you expect that it might affect results.
Lines 145-146: did the number of bees that did not consume the solution vary by treatment?
Lines 171-172: did the number of bees that did not respond here vary by treatment?
Lines 188-190: Wondering why you don’t look at binary response (0/1) (for a given bee) over the 15 trials, including ‘trial’ as a fixed factor (and bee as a random factor) using a GLMM with a binomial distribution? It seems like you might get more information out of your data this way, rather than reducing it to the variables listed?
Line 191: shouldn’t bee size be a covariate here rather than a fixed factor?
Line 204: do you mean mg of pollen?
Line 233 section: I think it would be good to report how long each trial lasted and the inter-trial interval in this section if this isn’t reported elsewhere?

Reviewer 3 ·

Basic reporting

No comment.

Experimental design

No comment.

Validity of the findings

No comment.

Additional comments

It was delightful to read such interesting and well conceived manuscript. It is very well written and brings valuable information about potential impacts of a new insecticide on bees, the most important pollinators worldwide.

Reviewer 4 ·

Basic reporting

The manuscript, "No evidence for negative impacts of acute sulfoxaflor exposure on bee olfactory conditioning or working memory" by Siviter et al. addresses a very important issue regarding the sublethal effects of insecticides on olfactory conditioning performance and working memory performance of honeybees and bumblebees. The research was thorough and the findings were convincing. The paper wrote well and clearly. I think the manuscript can be published after minor revision.

Experimental design

More details regarding bumblebee colonies should be provided. These colonies are provided by a company, but what were the rearing conditions? These colonies are from the UK? Which localities?

Validity of the findings

The figures text could be in larger font size.

Additional comments

The title could express a curiosity regarding this investigation, and not include an affirmative sentence about the main evidence found.

For instance: There are negative impacts of acute sulfoxaflor exposure on bee olfactory conditioning or working memory?

Or

Negative impacts of acute sulfoxaflor exposure on bee olfactory conditioning or working memory

---

## Round 0.2 · accepted · Accept

Thank you for your efforts in addressing reviewer comments.

#